# Long-Term HBsAg Titer Kinetics with Entecavir/Tenofovir: Implications for Predicting Functional Cure and Low Levels

**DOI:** 10.3390/diagnostics14050495

**Published:** 2024-02-25

**Authors:** Soon Kyu Lee, Soon Woo Nam, Jeong Won Jang, Jung Hyun Kwon

**Affiliations:** 1Department of Internal Medicine, College of Medicine, Incheon St. Mary’s Hospital, The Catholic University of Korea, Seoul 21431, Republic of Korea; blackiqq@catholic.ac.kr (S.K.L.); drswnam@catholic.ac.kr (S.W.N.); 2The Catholic University Liver Research Center, College of Medicine, The Catholic University of Korea, Seoul 06591, Republic of Korea; garden@catholic.ac.kr; 3Department of Internal Medicine, College of Medicine, Seoul St. Mary’s Hospital, The Catholic University of Korea, Seoul 06591, Republic of Korea

**Keywords:** HBsAg, functional cure, entecavir, tenofovir, HBeAg

## Abstract

The long-term kinetics of quantitative HBsAg levels in HBV-infected patients treated with entecavir or tenofovir, as well as the role of quantitative HBsAg in predicting functional cure (HBsAg loss) and low HBsAg levels (<2 log IU/mL) remain unclear. Of some 1661 consecutively enrolled patients newly treated with entecavir or tenofovir, we analyzed 852 patients who underwent serial HBsAg level checks every 6–12 months. The primary outcomes included long-term kinetics in HBsAg levels and the rate of functional cure and achieving low HBsAg levels. Over a mean 6.3-year follow-up, the functional cure rate was 2.28% (n = 19), and 12.9% (n = 108) achieved low HBsAg levels. A significant HBsAg level reduction was seen in the first treatment year (*p* < 0.05), with another stepwise decrease between year 6–7. These trends were pronounced in patients with chronic hepatitis and HBeAg-positivity compared to those with cirrhosis and HBeAg-negativity, respectively. Baseline HBsAg of ≤3 log IU/mL and the first-year HBsAg reduction were key predictors for both functional cure and low HBsAg levels (*p* < 0.05). In conclusion, our findings elucidate the stepwise reduction in quantitative HBsAg dynamics during high-potency NA therapy (entecavir or tenofovir) along with variations based on different conditions. We also underscore the significance of quantitative HBsAg titer in predicting functional cure and low-HBsAg levels.

## 1. Introduction

Chronic hepatitis B (CHB) virus infection, affecting an alarming 296 million individuals as of 2019, stands as a primary cause of hepatic decompensation and hepatocellular carcinoma (HCC) [1,2]. With the advancements in antiviral therapies, particularly the advent of high potency nucleos(t)ide analogues (NAs) like entecavir and tenofovir, the risk of HCC has dramatically diminished, consequently bolstering the survival rates of CHB patients [3]. Nevertheless, even though these potent NAs efficiently suppress HBV DNA levels to undetectable levels, the rate of hepatitis B surface antigen (HBsAg) loss remains modest, with an annual rate of only 0.3–0.8%, and most patients will probably need decades of therapy to achieve HBsAg loss [4,5,6].

HBsAg loss, termed a “functional cure”, denotes the elimination of infection with low levels of cccDNA remaining, establishing it as a surrogate endpoint for NA treatment [7,8,9]. Given its correlation with favorable clinical outcomes, there has been a dedicated effort to achieve HBsAg loss [7,8,9]. Intriguingly, studies indicate that NA responders with low HBsAg levels (<2logIU/mL) at the end-of-treatment (EOT) have a 5-year rate HBsAg loss rate exceeding 30%, suggesting a potential cut-off level and predictor of HBsAg loss [9]. Hence, to achieve functional cure, accurately predicting HBsAg loss and low HBsAg levels are essential for CHB patients following NA treatment.

Quantitative HBsAg, which detects Dane particle and sub-viral particles, is known to positively correlate with intrahepatic cccDNA, suggesting that HBsAg kinetics could act as an indicator for HBsAg loss [7]. Notably, both baseline and 12 month HBsAg levels have been proposed as predictors for HBsAg loss [9,10,11]. Despite these findings, the long-term kinetics of HBsAg over 5 years remains unclear. It is yet to be determined whether the slow decline of HBs levels, even with high-potency NAs, will follow a linear or stepwise pattern. Furthermore, in light of prior data that showed a sharper decrease in HBsAg levels with peginterferon than with entecavir therapy [12], it is essential to delve deeper into the kinetics variations and their related impacts on HBsAg loss according the NA drugs, either entecavir or tenofovir, in the current era of high-potency antiviral treatments.

To address these issues, our study engaged a sizable cohort with extensive longitudinal data to assess the long-term kinetics of HBsAg based on various disease status, such as HBeAg status, presence of cirrhosis, and the specific NA type, in CHB patients treated with either entecavir or tenofovir. Furthermore, we sought to elucidate the role of quantitative HBsAg levels in forecasting clinical outcomes, encompassing functional cure, low HBsAg levels, and HCC development.

## 2. Materials and Methods

### 2.1. Study Population

The study population included a total of 1661 treatment-naïve HBV-infected patients diagnosed with chronic hepatitis or liver cirrhosis (LC), newly initiated with antiviral therapy with entecavir or tenofovir at the liver unit of Incheon St. Mary’s Hospital, a tertiary university hospital, between 2006 and 2020. The initiation of NA treatment adhered to the appropriate HBV treatment guidelines in Korea [3]. These patients, who had no significant history of alcohol consumption or fatty liver diseases, were consecutively screened for study eligibility. Among them, 809 were excluded for the following reasons: they underwent liver transplantation (n = 39), their absence of baseline HBsAg levels (n = 640), or they were lost to follow-up within 12 months (n = 130). Consequently, 852 patients (entecavir, n = 287; tenofovir, n = 565) were included in the analysis of HBsAg level kinetics (Appendix A). This study adhered to the principles of the Declaration of the Helsinki and received approval from the Institutional Review Board of the Incheon St. Mary’s Hospital, The Catholic University of Korea (XC20WIDI0173).

### 2.2. Clinical and Laboratory Data

Clinical data were collected at the time of initiating NA treatment and included age, sex, and the presence of LC. Baseline laboratory variables included aspartate transaminase (AST), alanine transaminase (ALT), bilirubin, creatinine, albumin, platelet counts, white blood cell counts, international normalized ratio (INR), HBV DNA levels, HBeAg, and quantitative HBsAg levels. The presence of LC was diagnosed using methods such as liver biopsy, abdominal ultrasonography (US), computed tomography (CT), magnetic resonance imaging (MRI), or fibroscan. HBV DNA levels were ascertained using a real-time polymerase chain reaction (PCR) assay (Abbott Laboratories, Abbott Park, IL, USA), with a detection threshold of 10 IU/mL. Quantitative HBsAg level measurements were performed using commercial immunoassays (Abbott Laboratories, Abbott Park, IL, USA). HbeAg levels were also quantified using the Architect HbeAg assay (Abbott Laboratories, Abbott Park, IL, USA). The assay results were validated with a reference standard (100 PEIU/mL) obtained from the Paul Ehrlich Institute (Langen, Germany) [13].

### 2.3. Follow-Up

Patients underwent regular follow-ups every 3–6 months, during which laboratory tests such as liver function tests, complete blood count, HbeAg/Ab, and HBV DNA were conducted. The quantitative HbsAg level was also monitored every 6–12 months after initiating NA therapies. Additionally, abdominal US and alpha-fetoprotein (AFP) tests were performed every 6 months for HCC surveillance, in accordance with the treatment guidelines [3,7]. The diagnosis of HCC was confirmed based on histologic and/or imaging results from dynamic CT or contrast-enhanced MRI, characterized by arterial phase hypervascularity followed by washout in the venous or delayed phase [14].

### 2.4. Outcome Assessment

The primary outcome of the study was to analyze the long-term kinetics of HbsAg levels, coupled with identifying the rate of functional cure and the achievement of low HBsAg levels during NA therapy. A functional cure, or HBsAg loss, was defined as the sustained undetectability of HBV DNA levels in conjunction with the absence of serum HBsAg regardless of HBsAb status. A low HBsAg level referred to an achievement of an HBsAg level below 2 logIU/mL during the follow-up.

Secondary outcomes comprised the comparison of HBsAg level kinetics based on the presence of cirrhosis, HBeAg status, and specific NA types in CHB patients during either entecavir or tenofovir treatment. Furthermore, we aimed to pinpoint predictive factors for achieving a functional cure and a low HBsAg level, as well as to evaluate the incidence of HCC development during follow-up after initiating NA treatment.

### 2.5. Statistical Analysis

Baseline characteristics are presented as median (interquartile range [IQR]) or mean ± standard deviation for continuous variables, and as counts (percentage) for categorical variables. Comparisons between groups for continuous variables were conducted using Student’s t-test or the Mann–Whitney U test, depending on data distribution, and the Chi-square test or Fisher’s exact test were used for categorical variables, as deemed appropriate. Kaplan–Meier analysis was employed to estimate rates of functional cure and the achievement of low HBsAg levels. Significant factors for functional cure and a low HBsAg levels were identified using a Cox regression analysis. A two-sided *p* value of <0.05 was considered significant. All statistical analyses were carried out using R software (version 4.3.1; http://carn.r-project.org accessed on 1 July 2023).

## 3. Results

### 3.1. Baseline Characteristics

Of the included 852 patients, the median age was 51.0 years (IQR, 43.5–58.0), and 517 (60.7%) were male. Approximately half (n = 415, 48.7%) had LC at the time of initiating NA therapy. A total of 498 patients (58.5%) tested positive for HBeAg at baseline, with the median HBV level of the entire cohort being 6.6 logIU/mL. Median AST and ALT levels at baseline stood at 76.0 and 83.0 IU/mL, respectively. The median HBsAg level was recorded at 2618.0 IU/mL (IQR, 1167.0–5523.4) (Table 1).

### 3.2. Long-Term Kinetics in HBsAg Levels following NA Therapy

Over a mean follow-up of 6.3 ± 3.6 years, HBsAg levels exhibited a significant reduction in the first year following NA treatment (*p* < 0.05). Subsequently, HBsAg levels continued to decrease, showcasing a second stepwise reduction between the 6th and 7th years post-treatment (Figure 1A). For a more granular analysis, we categorized patients into three distinct groups based on their HBsAg levels: the low-HBsAg group (HBsAg < 2 log IU/mL), the intermediate-HBsAg group (2 log ≤ HBsAg < 3 log IU/mL), and the high-HBsAg group (HBsAg ≥ 3 log IU/mL). As depicted in Figure 1B, the proportions of patients in the low- and intermediate-HBsAg groups increased from 6.5% and 15.3% at baseline to 7.7% and 17.9% after the first year, respectively. By the 8th year of entecavir or tenofovir treatment, these rates further escalated to 14.2% and 39.6% for the low and intermediate groups, respectively.

### 3.3. Achievement of Functional Cure and Low HBsAg Levels

We subsequently assessed the rates of functional cure and entry into the low-HBsAg group. Our findings revealed a functional cure rate of 2.28% (n = 19; 0.34/100 person-years [PY]), while the low-HBsAg group was achieved by 108 patients (12.9%; 1.95/100 PY). In-depth observation showed a pronounced HBsAg reduction during the first treatment year both for those achieving functional cure and those not, with a steeper decline in the latter (Appendix A). A comparable pattern also emerged in the intermediate-to-high HBsAg and low-HBsAg groups (Appendix A). This once again underscored a significant drop in HBsAg during the first year of therapy, followed by a 2nd stepwise decline between the 6th and 7th years post-treatment.

Detailed characteristics and outcomes for patients attaining functional cure are presented in Appendix A. Out of the 19 patients (average age: 53.5 years), 13 (68.4%) were male, while 8 (42.1%) had LC at baseline. The distribution between the two treatments—entecavir (n = 9; 48.4%) and tenofovir (n = 10; 52.6%)—did not significantly differ in terms of achieving functional cure. On average, functional cure was achieved within 3.68 years. Importantly, none of the patients in the functional cure group developed HCC, while three exhibited disease regression. However, in the non-functional cure group, 87 patients developed HCC without statistical significance compared to the functional cure group.

### 3.4. Changes in HBsAg Levels Based on Various Disease Status

To further delve into the HBsAg level kinetics post-NA therapy, we segmented our analysis based on HBeAg status and the presence of LC. It was observed that HBeAg-positive patients had notably higher baseline HBsAg levels and experienced a considerable reduction in the first year of therapy (*p* = 0.004), a trend that was not mirrored in HBeAg-negative patients (Figure 2A). After the initial year of therapy, a gradual decline in HBsAg levels was noted in both groups.

A similar analysis carried out with respect to LC presence revealed that chronic hepatitis patients, those without LC, not only had higher baseline HBsAg levels but also displayed a more significant decrease during the first year of therapy (*p* = 0.005) compared to patients with LC (Figure 2B). Additionally, a secondary stepwise reduction was noticeable after 5 and 6 years of NA therapy in patients with chronic hepatitis and cirrhosis, respectively.

### 3.5. Changes in HBsAg Levels among HBeAg (+) Patients

Considering that HBeAg-positive patients presented with a higher baseline HBsAg level coupled with a swift decline during NA treatments, we further investigated changes in HBsAg levels specifically among this group. Chronic hepatitis patients maintained a notable reduction in HBsAg levels during the first year of therapy (*p* < 0.001), followed by a secondary stepwise reduction between the 5th and 7th years of treatment (Figure 2C). Conversely, the trajectory for LC patients demonstrated a more gradual decline throughout their treatments.

### 3.6. Comparison of HBsAg Level Changes between Tenofovir and Entecavir

When evaluating the influence of specific NAs used, we observed no significant difference in baseline HBsAg levels between the entecavir and tenofovir groups. Notably, their reduction patterns differed: the tenofovir group showed a pronounced decrease in the first year (*p* = 0.005), whereas the entecavir group underwent a swift reduction between the first and second year of treatment (*p* = 0.019) (Figure 2D). Both groups presented a consistent decrease in HBsAg levels throughout the subsequent follow-up periods, resulting in no discernible difference in HBsAg decline between the two NAs after 7–8 years of treatment.

Further analysis on HBsAg level variations based on the type of NAs given to HBeAg-positive patients revealed patterns consistent with our initial observations. In the tenofovir group, a sharp reduction was evident during the first therapy year (*p* = 0.005), followed by a more gradual decrease thereafter (Figure 2E). In contrast, the entecavir group experienced a substantial drop between the first and second treatment years, along with a milder reduction during the initial year—mirroring trends seen in the entire population.

### 3.7. Factors Associated with Functional Cure and Low HBsAg Levels

Considering the observed differences in the baseline HBsAg levels between the functional cure and non-functional cure groups, we first assessed the impact of baseline HBsAg levels on achieving functional cure and low HBsAg levels. In our analysis, ‘HBsAg reduction at 1 year’ was defined as the decrease in HBsAg levels at the 1-year mark relative to baseline levels. Patients with HBsAg ≤ 3 log IU/mL exhibited a significantly higher rate of both achieving functional cure and low HBsAg levels (<2log IU/mL) (*p* < 0.001 for both; Figure 3). Meanwhile, no significant differences in the cumulative incidence rate for the achievement of functional cure and low HBsAg levels were observed between the tenofovir and entecavir groups (Appendix A).

Building on these findings which underscore the importance of baseline HBsAg levels and HBsAg reduction during the first year of therapy, we further analyzed their influence on functional cure and low HBsAg through a Cox regression analysis. Both baseline HBsAg ≤ 3 log IU/mL and a reduction in HBsAg at one year emerged as significant determinants for achieving functional cure (*p* < 0.001 and *p* = 0.010, respectively). These factors also significantly influenced the achievement of low HBsAg levels (*p* < 0.001 for both) (Table 2), thereby emphasizing their predictive value for both functional cure and low HBsAg levels.

To further investigate the impact of baseline HBsAg levels and HBsAg reduction during the first year of therapy, we conducted a 5-year landmark analysis (Appendix A). Baseline HBsAg ≤ 3 log IU/mL remained a significant factor for achieving both functional cure and low HBsAg level (*p* < 0.001 for both). Additionally, a reduction in HBsAg at one year was also a significant factor for achieving low HBsAg (*p* < 0.001). These results highlight the enduring significance of baseline HBsAg levels and HBsAg reduction during the first year in predicting both functional cure and low HBsAg loss.

## 4. Discussion

This comprehensive HBV cohort study elucidated HBsAg kinetics across diverse disease status after initiating high-potency NAs. HBsAg levels markedly reduced in the first treatment year, followed by a second stepwise decrease between the 6th and 7th years. These patterns were more distinctly observed in HBeAg-positive patients with chronic hepatitis compared to LC and HBeAg-negative patients. Reduction patterns diverged between entecavir and tenofovir, with the latter showing a swifter decrease in the first year of treatment. Importantly, a baseline HBsAg of ≤3 log IU/mL and a first-year reduction in HBsAg emerged as critical predictors for both functional cure and low HBsAg levels. This study highlights the variance in HBsAg changes based on disease status and NA type along with underscoring its predictive role for patients’ outcomes.

Through an extended analysis of HBsAg kinetics, we identified distinct patterns in HBsAg level changes: a rapid decline in the first year, followed by a stepwise reduction between the 6th and 7th years of treatment. The significant drop during the first year of high-potency NA treatment aligns with findings from previous studies [15,16]. However, the limited follow-up duration in earlier research introduced uncertainties regarding long-term HBsAg kinetics, especially beyond 5 years. Our long-term analysis, for the first time, highlighted this stepwise decrease between the 6th and 7th treatment years, indicating that the decline of HBsAg titer is not linear. This suggests that achieving HBsAg loss may require less time than previously proposed [4,5,6]. These findings are pivotal and warrant validation in future studies, particularly those involving the treatment with tenofovir alafenamide.

Indeed, the observed patterns were more pronounced in patients with chronic hepatitis and those who were HBeAg-positive compared to those with LC and HBeAg-negative status. In relation to HBeAg-positivity, several studies have reported a gradual decrease in HBsAg levels at a rate of −0.080 log IU/mL per year in HBeAg-negative patients [10,17,18], consistent with our findings. Prior research has indicated that this slow reduction is consistent across both low-potency and high-potency drugs, including entecavir and tenofovir [9,10]. Given the natural progression of chronic HBV infection [9,19], it is expected for patients with cirrhosis to exhibit low HBsAg levels. Further, our study revealed that the HBsAg levels, which are correlated with intrahepatic cccDNA levels [20], diminish gradually during prolonged NA treatments in cirrhotic patients.

Leveraging the benefits of long-term analysis, our study recorded a functional cure rate of 2.28%, translating to 0.34/100 PY. This rate is significantly lower than a previous study that assessed HBsAg kinetics in CHB patients treated with tenofovir, where they observed an 8.6% functional cure rate with up to 5 years [16]. The diminished functional cure rate in our study may stem from the prevalence of genotype C in Korea [21], which has been identified as an independent risk factor for HCC and liver cirrhosis [22]. In fact, a Korean study reported an annual functional cure rate of 0.3–0.4% both with and without NA treatment, aligning closely with our findings [3,23]. Considering the predictive significance of low HBsAg levels for achieving future functional cures [24], we also assessed the rate of low HBsAg levels within our longitudinal cohort, arriving at a result of 1.95/100 PY. While our study found a modest prevalence of low HBsAg levels following NA treatments, patients with such levels might contemplate stopping NAs [9], given the heightened likelihood of achieving a functional cure after discontinuing NAs [9,25].

A salient finding in our study was the swifter decline of HBsAg levels in patients treated with tenofovir compared to those with entecavir. Notably, the tenofovir group displayed a pronounced reduction within the first year of treatment, while the entecavir group exhibited a sharp decline between the first and second treatment years. Past research indicated that HBsAg levels did not decrease significantly with entecavir treatment compared to peginterferon therapy [12]. The slower reduction in HBsAg levels during entecavir therapy was also corroborated by another study with a two-year follow-up [26]. On the other hand, the HBsAg kinetics associated with tenofovir treatment showed a rapid early drop post-initiation [16]. Consequently, our data indicates that tenofovir therapy might be more adept at rapidly reducing HBsAg levels, which implies enhanced control of the host’s immune response against the virus and a decrease in cccDNA [27]. The enhanced efficacy of tenofovir may be linked to its additional mechanisms compared to entecavir, notably the inhibition of IL-10 and Akt phosphorylation, alongside the induction of IL-12p70 [28,29,30]. However, since no differences were observed in the functional cure rate between tenofovir and entecavir therapies, which was in-line with a previous study [31], further studies evaluating the long-term HBsAg kinetics and functional cure rates between these potent NAs are warranted.

Our study underscores the predictive value of HBsAg levels for both functional cure and low HBsAg levels. Specifically, a baseline HBsAg of ≤3 log IU/mL and a reduction in HBsAg during the first year emerged as significant indicators for these outcomes during prolonged NA therapy. These findings align with several previous studies highlighting the importance of baseline HBsAg levels and its first-year decline in predicting functional cure [11,27,32]. Moreover, we emphasize the significance of these markers in achieving low HBsAg levels. Given that reduced HBsAg levels indicate improved host immune control, it is logical that a first-year reduction in HBsAg correlates with the achievement of low HBsAg levels. Thus, monitoring HBsAg levels at baseline and after first-year of high-potency NA treatment is crucial for forecasting clinical outcomes, including functional cure and achieving low HBsAg levels.

Our study presents several limitations. Firstly, it is retrospective in nature. Secondly, as our study was conducted in a single center located in a region predominantly characterized by genotype C, there is a need for verification of our findings in other regions with other genotypes, particularly B or D. This is important as different genotypes may contribute to varying clinical outcomes, thereby affecting the implications of monitoring HBsAg levels for patient outcomes. However, despite these constraints, our research leveraged a large-scale, long-term cohort, offering valuable insights into quantitative HBsAg kinetics during entecavir or tenofovir treatments. Our detailed analysis included comparative analyses based on HBeAg positivity and the presence of LC. Thirdly, our study did not evaluate HBV RNA levels, and we were unable to perform liver biopsies to directly quantify cccDNA and HBsAg concentrations. Fourthly, a significant percentage of patients achieving functional cure had LC. Although a functional cure typically suggests a favorable prognosis, caution is advised regarding the discontinuation of NAs in patients with cirrhosis; continuation of NA treatment is generally recommended. Therefore, further studies in diverse countries with varying baseline characteristics are warranted to reinforce our findings.

In conclusion, our study highlighted a significant reduction in quantitative HBsAg levels during the first year of treatment, succeeded by a second stepwise decrease between 6–7 years in patients treated with entecavir and tenofovir. This rapid decline in the initial treatment year was more pronounced in patients with chronic hepatitis, HBeAg-positive status, and those undergoing tenofovir treatment. Moreover, a baseline HBsAg of ≤3 log IU/mL and a first-year reduction in HBsAg emerged as pivotal determinants for both functional cure and achieving low HBsAg levels. Taken together, our results provide insights into the variance in HBsAg kinetics under various conditions and highlight its role in forecasting patient outcomes.

## Figures and Tables

**Figure 1 diagnostics-14-00495-f001:**
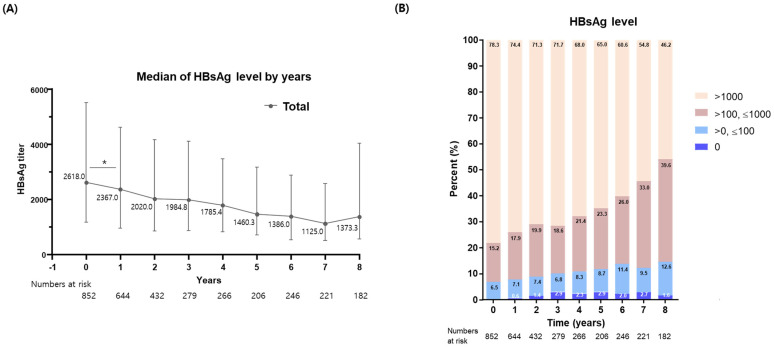
Serial changes in (**A**) the HBsAg levels and (**B**) their proportion of groups after nucleos(t)ide analogue therapy. *, *p* < 0.05.

**Figure 2 diagnostics-14-00495-f002:**
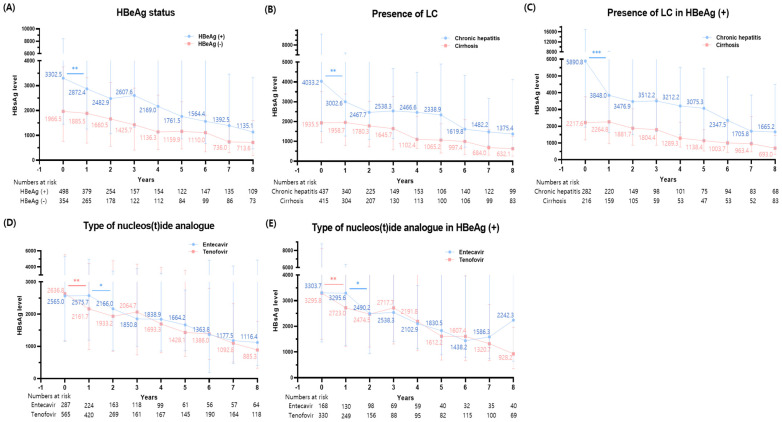
Serial changes in the HBsAg levels according to (**A**) HBeAg status, (**B**) the presence of liver cirrhosis in entire population, (**C**) HbeAg—positive patients, and the type of nucleos(t)ide analogue therapy in (**D**) the entire population and (**E**) HBeAg—positive patients. *, *p* < 0.05; **, *p* < 0.01; ***, *p* < 0.001.

**Figure 3 diagnostics-14-00495-f003:**
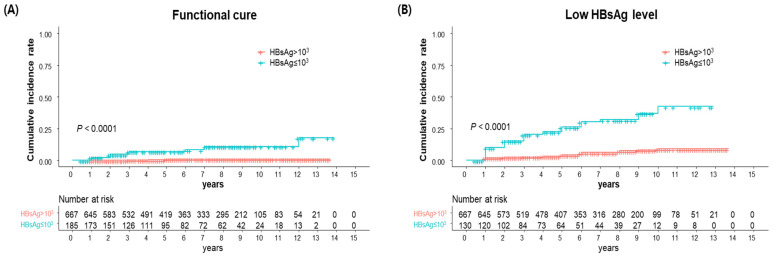
Kaplan–Meier curve for (**A**) functional cure and (**B**) low HBsAg level according to baseline HBsAg level.

**Table 1 diagnostics-14-00495-t001:** Baseline characteristics of entire population.

Variables	Total (N = 852)	Achievement of Functional Cure			Achievement of Low HBsAg Titer (≤100 IU/mL)		
		Non-Functional Cure	Functional Cure	*p*-Value	High HBsAg (>100)	Low HBsAg (≤100)	*p*-Value
		(n = 833)	(n = 19)		(n = 725)	(n = 127)	
Age, years	51.0 [43.5; 58.0]	51.0 [43.0; 58.0]	55.0 [47.0; 63.0]	0.216	49.7 ± 11.4	55.7 ± 12.3	<0.001
Male	517 (60.7%)	504 (60.5%)	13 (68.4%)	0.645	427 (58.9%)	90 (70.9%)	0.014
Cirrhosis	415 (48.7%)	407 (48.9%)	8 (42.1%)	0.726	347 (47.9%)	68 (53.5%)	0.278
HBeAg positivity	498 (58.5%)	488 (58.6%)	10 (52.6%)	0.776	441 (60.8%)	57 (44.9%)	0.001
HBVDNA, log IU/mL	6.6 [5.5; 7.5]	6.6 [5.5; 7.5]	6.3 [5.4; 6.8]	0.126	6.7 [5.6; 7.6]	6.1 [5.1; 7.1]	<0.001
AST, IU/mL	76.0 [49.0; 140.5]	76.0 [49.0; 139.0]	94.0 [43.0; 270.0]	0.644	152.8 ± 288.6	199.2 ± 346.5	0.156
ALT, IU/mL	83.0 [44.5; 178.5]	83.0 [44.0; 175.0]	153.0 [60.5; 449.5]	0.108	177.5 ± 295.1	241.0 ± 473.2	0.145
Tbil, mg/dL	1.0 [0.7; 1.4]	1.0 [0.7; 1.4]	1.2 [0.8; 2.1]	0.321	1.6 ± 2.7	2.0 ± 2.6	0.099
Alb, g/dL	4.0 [3.6; 4.3]	4.0 [3.6; 4.3]	4.3 [4.0; 4.5]	0.009	3.9 ± 0.6	3.8 ± 0.7	0.062
Cr, mg/dL	0.8 [0.7; 0.9]	0.8 [0.7; 0.9]	0.8 [0.8; 0.9]	0.201	0.9 ± 0.9	0.8 ± 0.2	0.270
INR	1.1 [1.1; 1.2]	1.1 [1.1; 1.2]	1.1 [1.1; 1.1]	0.251	1.2 ± 0.3	1.2 ± 0.3	0.096
Plt, 103/μL	149.0 [107.0; 191.0]	149.0 [107.0; 191.0]	159.0 [141.5; 191.5]	0.405	156.4 ± 62.6	140.3 ± 64.6	0.008
WBC, μL	5190.0 [4100.0; 6425.0]	5190.0 [4100.0; 6420.0]	5220.0 [4500.0; 6800.0]	0.496	5504.4 ± 2173.9	5585.5 ± 2170.0	0.698
CTP score	5.0 [5.0; 6.0]	5.0 [5.0; 6.0]	5.0 [5.0; 5.0]	0.659	5.7 ± 1.5	5.9 ± 1.7	0.183
MELD score	6.0 [6.0; 8.0]	6.0 [6.0; 8.0]	6.0 [6.0; 9.0]	0.536	7.7 ± 3.8	8.5 ± 4.2	0.030
HBsAg, IU/mL	2618.0 [1167.0; 5523.4]	2650.4 [1209.5; 5636.1]	71.7 [22.9; 246.6]	<0.001	24,832.4 ± 366,366.3	1711.6 ± 4953.7	0.090

**Table 2 diagnostics-14-00495-t002:** Univariate and multivariate Cox regression analysis for functional cure and low HBsAg level.

Functional Cure							Low HBsAg Level (<2 log IU/mL)						
Variables	Univariate Analysis			Multivariable Analysis			Variables	Univariate Analysis			Multivariable Analysis		
	HR	95% CI	*p*-Value	HR	95% CI	*p*-Value		HR	95% CI	*p*-Value	HR	95% CI	*p*-Value
Age	1.03	0.99, 1.07	0.2				Age	1.02	1.00, 1.05	0.041	1.01	0.99, 1.04	0.4
Male	1.48	0.56, 3.90	0.4				Male	2.13	1.25, 3.63	0.006	1.85	1.01, 3.40	0.048
HBeAg (+)	0.73	0.30, 1.80	0.5				HBeAg (+)	0.67	0.42, 1.06	0.087	0.88	0.50, 1.55	0.7
HBVDNA	1.00	1.00, 1.00	0.15				HBVDNA	1.00	1.00, 1.00	0.020	1.00	1.00, 1.00	0.10
Alb	4.01	1.26, 12.7	0.018	6.08	1.59, 23.3	0.008	Alb	0.66	0.47, 0.94	0.021	0.67	0.41, 1.08	0.10
Plt	1.00	1.00, 1.01	0.6				Plt	1.00	0.99, 1.00	0.11			
AST	1.00	1.00, 1.00	<0.001	1.00	1.00, 1.00	0.8	AST	1.00	1.00, 1.00	0.030	1.00	1.00, 1.00	>0.9
ALT	1.00	1.00, 1.00	<0.001	1.00	1.00, 1.00	0.079	ALT	1.00	1.00, 1.00	0.046	1.00	1.00, 1.00	0.3
Tbil	1.06	0.97, 1.16	0.2				Tbil	1.05	1.00, 1.11	0.064	0.99	0.87, 1.12	0.8
INR	0.44	0.03, 6.55	0.6				INR	1.16	0.81, 1.68	0.4			
Cr	0.97	0.40, 2.35	>0.9				Cr	0.98	0.66, 1.46	>0.9			
MELD	1.03	0.96, 1.11	0.4				MELD	1.03	1.00, 1.07	0.060	0.98	0.88, 1.10	0.8
Antiviral	0.71	0.28, 1.80	0.5				Antiviral	1.51	0.90, 2.54	0.12			
Cirrhosis	0.84	0.34, 2.09	0.7				Cirrhosis	1.03	0.65, 1.64	0.9			
HBsAg ≤ 10^3^ at baseline	15.5	5.15, 46.8	<0.001	21.6	7.00, 66.7	<0.001	HBsAg ≤ 10^3^ at baseline *	5.94	3.74, 9.44	<0.001	6.37	3.70, 10.9	<0.001
HBsAg reduction at 1 yr	3.17	0.92, 10.9	0.066	5.14	1.47, 18.0	0.010	HBsAg reduction at 1 yr	1.86	1.04, 3.33	0.036	2.82	1.54, 5.17	<0.001

HR = Hazard Ratio, CI = Confidence Interval, * HBsAg ≤ 100 IU/mL at baseline were excluded.

## Data Availability

The data presented in this study are available on request from the corresponding author.

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
