# Peer review of "Long-Term HBsAg Titer Kinetics with Entecavir/Tenofovir: Implications for Predicting Functional Cure and Low Levels"

_diagnostics, 2024, doi:10.3390/diagnostics14050495_

Round 1

Reviewer 1 Report

Comments and Suggestions for Authors

In this study, authors analyzed the long-term HBsAg titer kinetics in HBV-infected patients treated with entecavir or tenofovir. The topic is meaningful and the results are presented clearly. However, some issues are suggested to be dealt with.

1.         The topic of this study is very interesting. However, several similar studies have been published (doi: 10.1093/infdis/jiab241; 10.1111/jvh.13306; 10.1007/s12072-022-10411-x). What is the highlight of the current study?

2.         It is recommended that authors should list all the details of assays used in the measurement of laboratory data, like the serum HBeAg and quantitative HBsAg.

3.         What is the definition of functional cure? Please provide the references. It has been reported that functional cure of hepatitis B is defined as sustained undetectable circulating HBsAg and HBV DNA after a finite course of treatment (doi: 10.1016/j.jhep.2021.11.024). Thus, should HBV DNA also be mentioned in this study? Why do authors only analyze the HBsAg?

4.         The quality of figures should be improved.

Comments on the Quality of English Language

Minor editing of English language required

Author Response

21 Feb, 2024

Professor Andreas Kjaer.
Editor-in-Chief

Diagnostics

Thank you for your consideration of the manuscript entitled “Long-term HBsAg titer kinetics with entecavir/tenofovir: Implications for predicting functional cure and low levelsfor the publication in Diagnostics.

We have carefully considered the suggestions by the reviewers. We fully agree with the opinions of the reviewers and have revised the manuscript accordingly. The responses to the reviewers are appended by point-to-point in the revised version and we hope that our explanations and revisions are satisfactory. Point-to-point answers are provided from the next page of this cover letter.

Point-by-Point Response to Reviewer I

In this study, authors analyzed the long-term HBsAg titer kinetics in HBV-infected patients treated with entecavir or tenofovir. The topic is meaningful and the results are presented clearly. However, some issues are suggested to be dealt with.

Q1) The topic of this study is very interesting. However, several similar studies have been published (doi: 10.1093/infdis/jiab241; 10.1111/jvh.13306; 10.1007/s12072-022-10411-x). What is the highlight of the current study?

Ans) We appreciate the reviewer’s insightful comment. Indeed, while there have been several studies related to quantitative HBsAg or functional cure, our study addresses certain limitations observed in these previous works. Specifically: First study referenced with doi:10.1093/infdis/jiab241 did not analyze the serial changes in quantitative HBsAg levels. Second study at 10.1111/jvh.13306 was limited to patients treated with entecavir only. Third study at 10.1007/s12072-022-10411-x did not analyze quantitative HBsAg titer and its changes. Therefore, our study's strength lies in its comprehensive assessment of dynamic two phasic changes in quantitative HBsAg titers and their clinical implications for functional cure and low HBsAg titer in NUC treated HBV-infected patients, who already showed undetectable serum HBV DNA. This includes a diverse clinical status spectrum and treatments with both entecavir and tenofovir. Following the reviewer’s suggestion, we have added the following highlights and strengths to our revised manuscript:

“In conclusion, our findings elucidate the stepwise reduction in quantitative HBsAg dynamics during high-potency NA therapy (entecavir or tenofovir) along with variations based on different conditions. We also underscore the significance of quantitative HBsAg titer in predicting functional cure and low-HBsAg levels.” (On page 1)

“However, despite these constraints, our research leveraged a large-scale, long-term cohort, offering valuable insights into quantitative HBsAg kinetics during entecavir or tenofovir treatments. Our detailed analysis included comparative analyses based on HBeAg positivity and presence of LC.” (On page 9)

“In conclusion, our study highlighted a significant reduction in quantitative HBsAg levels during the first year of treatment, succeeded by a second stepwise decrease between 6-7 years in patients treated with entecavir and tenofovir.” (On page 9)

Q2) It is recommended that authors should list all the details of assays used in the measurement of laboratory data, like the serum HBeAg and quantitative HBsAg.

Ans) We would like to thank the reviewer’s suggestion regarding the detailing of assay methods. In response to the reviewer’s comments, we have included the details of assays used in the measurement of HBV DNA levels in the revised manuscript as follows:

“HBV DNA levels were ascertained using a real-time polymerase chain reaction (PCR) assay (Abbott Laboratories, Abbott Park, IL), with a detection threshold of 10 IU/mL. Quantitative HBsAg level measurements were performed using commercial immunoassays (Abbott Laboratories, Abbott Park, IL). HBeAg levels were also quantified using the Architect HBeAg assay (Abbott Laboratories, Abbott Park, IL), The assay results were validated with a reference standard (100 PEIU/mL) obtained from the Paul Ehrlich Institute (Langen, Germany) [13].” (On page 2)

Q3) What is the definition of functional cure? Please provide the references. It has been reported that functional cure of hepatitis B is defined as sustained undetectable circulating HBsAg and HBV DNA after a finite course of treatment (doi: 10.1016/j.jhep.2021.11.024). Thus, should HBV DNA also be mentioned in this study? Why do authors only analyze the HBsAg?

Ans) We are grateful to the valuable comments of the reviewer and apologize for any confusion regarding the definition of functional cure in our study. In line with the cited reference (10.1016/j.jhep.2021.11.024), functional cure in our research was defined as sustained undetectable levels of both HBsAg and HBV DNA. Since patients treated with entecavir and tenofovir typically exhibit undetectable HBV DNA levels, functional cure in these cases was not adequately predicted by HBV DNA levels alone. Hence, our study specifically focused on quantitative HBsAg levels as an indicative marker for predicting functional cure. In response to the reviewer’s comment, we revised the definition of functional cure in our manuscript to include the HBV DNA levels as follows:

“Nevertheless, even though these potent NAs efficiently suppress HBV DNA levels to undetectable levels, the rate of hepatitis B surface antigen (HBsAg) loss remains modest, with an annual rate of only 0.3-0.8% and most patients will probably need decades of therapy to achieve HBsAg loss [4-6]” (On page 1)

“A Functional cure, or HBsAg loss, was defined as the sustained undetectability of HBV DNA levels in conjunction with the absence of serum HBsAg regardless of HBsAb status. A low HBsAg level referred to an achievement of an HBsAg level below 2logIU/mL during the follow up.” (On page 3)

Q4) The quality of figures should be improved.

Ans) We thank the reviewer for their constructive feedback regarding the quality of the figures. In accordance with the reviewer's suggestion, we have improved the resolution of all figures in the revised manuscript to enhance their clarity and readability.

We hope that our revised manuscript will finally meet the qualification of publication in the Diagnostics.

We thank you for your time and efforts in dealing with our manuscript.

Sincerely yours,

Jung Hyun Kwon, M.D., Ph.D.,

Department of Internal Medicine, Incheon St. Mary's Hospital, College of Medicine, The Catholic University of Korea, #56 Dongsu-ro, Bupyeong-gu, Incheon, Republic of Korea

Tel. +82-32-280-7369; Fax. +82-32-280-5349; email: doctorkwon@catholic.ac.kr

Reviewer 2 Report

Comments and Suggestions for Authors

The manuscript by Lee SK et al reports long term results of NA therapy in CHB from Korea. The authors concentrate on HBsAg concentrations monitored during up to 7 years on either Tenofovir or Entecavir. 

1.    Disappointingly, there is no attempt to examine the main factors responsible for HBsAg production: HBV DNA in circulation and cccDNA in liver possibly reflected in HBV RNA levels. 

2.    At least for the few cases of functional cure, HBV DNA levels and HBeAg levels should be given in a separate table.

3.    Considering the relatively low frequency of functional cure and low HBsAg titer, the authors should try to compare their data with historical reports of untreated long term monitored cases in Korea where genotype C is nearly universal. 

4.    In the discussion, the known presentation of genotype C aggressive clinical outcome and high viral load might be mentioned and the possibility that NA therapy against cases infected with genotype B or D might be different.

5.    In the limitations of the study, the authors should indicate that liver biopsy were not part of the patient monitoring and that direct quantification overtime of cccDNA or liver tissue HBsAg concentrations could not be obtained and correlated to HBsAg levels.

6.    At the end of 3.3, it is mentioned that no HCC developed in the functional cure group. The authors should provide the data for HCC development in each group; it might be important to delineate the efficacy of NA in CHB.

Comments on the Quality of English Language

Adequate

Author Response

21 Feb, 2024

Professor Andreas Kjaer.
Editor-in-Chief

Diagnostics

Thank you for your consideration of the manuscript entitled “Long-term HBsAg titer kinetics with entecavir/tenofovir: Implications for predicting functional cure and low levelsfor the publication in Diagnostics.

We have carefully considered the suggestions by the reviewers. We fully agree with the opinions of the reviewers and have revised the manuscript accordingly. The responses to the reviewers are appended by point-to-point in the revised version and we hope that our explanations and revisions are satisfactory. Point-to-point answers are provided from the next page of this cover letter.

Point-by-Point Response to Reviewer II

The manuscript by Lee SK et al reports long term results of NA therapy in CHB from Korea. The authors concentrate on HBsAg concentrations monitored during up to 7 years on either Tenofovir or Entecavir.

Q1) Disappointingly, there is no attempt to examine the main factors responsible for HBsAg production: HBV DNA in circulation and cccDNA in liver possibly reflected in HBV RNA levels.

Ans) We sincerely appreciate the reviewer's insightful comments and concurred the reviewer’s comment.

In our study, functional cure is defined as sustained undetectable levels of both HBsAg and HBV DNA levels, in line with the previous research (10.1016/j.jhep.2021.11.024). To clarify, we have included commentary on HBV DNA levels in the definition of functional cure in our revised manuscript. Because HBV DNA levels are typically undetectable in patients treated with entecavir and tenofovir, functional cure could not be predictable by HBV DNA levels alone. Therefore, we our focus was primarily on quantitative HBsAg levels for predicting functional cure.

Regarding the HBV RNA levels, which as the reviewer rightly points out, play a crucial role in HBsAg production. Unfortunately, due to limitations in clinical accessibility and resources, we were unable to include an examination of HBV RNA levels in this study. This limitation has been noted and added to the discussion section of our revised manuscript.

Following the reviewer’s suggestions, we have made the necessary revisions to our manuscript to reflect these considerations and limitations as follows:

“Nevertheless, even though these potent NAs efficiently suppress HBV DNA levels to undetectable levels, the rate of hepatitis B surface antigen (HBsAg) loss remains modest, with an annual rate of only 0.3-0.8% and most patients will probably need decades of therapy to achieve HBsAg loss [4-6]” (On page 1)

“A Functional cure, or HBsAg loss, was defined as the sustained undetectability of HBV DNA levels in conjunction with the absence of serum HBsAg regardless of HBsAb status. A low HBsAg level referred to an achievement of an HBsAg level below 2logIU/mL during the follow up.” (On page 3)

“However, despite these constraints, our research leveraged a large-scale, long-term cohort, offering valuable insights into quantitative HBsAg kinetics during entecavir or tenofovir treatments Our detailed analysis included comparative analyses based on HBeAg positivity and presence of LC. Thirdly, our study did not evaluate HBV RNA levels, and we were unable to perform liver biopsies to directly quantify cccDNA and HBsAg concentrations.” (On page 9)

Q2) At least for the few cases of functional cure, HBV DNA levels and HBeAg levels should be given in a separate table.

Ans) We would like to thank the valuable comment of the reviewer. In the Supplementary Table 1, we have already detailed the characteristics of patients who achieved functional cure, including their HBV DNA levels and HBeAg status. Following the reviewer’s suggestion, we have further supplemented Supplementary Table 1 with additional information on HBeAg levels as follows:

Q3) Considering the relatively low frequency of functional cure and low HBsAg titer, the authors should try to compare their data with historical reports of untreated long term monitored cases in Korea where genotype C is nearly universal.

Ans) We are grateful to the reviewer’s insightful comments. In our discussion section, we previously noted the lower rate of functional cure in our study, which may be attributed to the high prevalence of genotype C in Korea. Following the reviewer’s comment, we have now included data on the functional cure rate in untreated patients in Korea, which is approximately 0.4% per year.  

“In fact, a Korean study reported an annual functional cure rate of 0.3-0.4% in both with and without NAs treatment, aligning closely with our findings [3,23].” (On page 8)

Q4) In the discussion, the known presentation of genotype C aggressive clinical outcome and high viral load might be mentioned and the possibility that NA therapy against cases infected with genotype B or D might be different.

Ans) We appreciate the comments of the reviewer. In the discussion section, we mentioned that our research was conducted at a single center in an area predominantly affected by genotype C and our findings should be verified in the other regions to solidify the implications of monitoring HBsAg levels for patient outcomes. Following the reviewer’s comment, we added that the results might be different in the area predominantly infected with genotype B or D as follows:

“Secondly, as our study was conducted in a single center located in a region predominantly characterized by genotype C, there is a need for verification of our findings in other regions with other genotypes, particularly B or D. This is important as different genotypes may contribute to varying clinical outcomes, thereby affecting the implications of monitoring HBsAg levels for patient outcomes.” (On page 9)

Q5) In the limitations of the study, the authors should indicate that liver biopsy were not part of the patient monitoring and that direct quantification overtime of cccDNA or liver tissue HBsAg concentrations could not be obtained and correlated to HBsAg levels.

Ans) We are grateful to the reviewer’s observation and totally agree with the comments. Following the reviewer’s comment, we have included the comments in the limitations section of our revised manuscript as follows:

“However, despite these constraints, our research leveraged a large-scale, long-term cohort, offering valuable insights into quantitative HBsAg kinetics during entecavir or tenofovir treatments Our detailed analysis included comparative analyses based on HBeAg positivity and presence of LC. Thirdly, our study did not evaluate HBV RNA levels, and we were unable to perform liver biopsies to directly quantify cccDNA and HBsAg concentrations.” (On page 9)

Q6) At the end of 3.3, it is mentioned that no HCC developed in the functional cure group. The authors should provide the data for HCC development in each group; it might be important to delineate the efficacy of NA in CHB.

Ans) We are thankful for the reviewer’s valuable comment and have accordingly revised manuscript to include more detailed data on HCC development according to the achievement of functional cure as follows:

“On average, functional cure was achieved within 3.68 years. Importantly, none of the patients in the functional cure group developed HCC, while three exhibited disease regression. However, in the non-functional cure group, 87 patients developed HCC without statistical significance compared to the functional cure group.” (On page 5)

We hope that our revised manuscript will finally meet the qualification of publication in the Diagnostics.

We thank you for your time and efforts in dealing with our manuscript.

Sincerely yours,

Jung Hyun Kwon, M.D., Ph.D.,

Department of Internal Medicine, Incheon St. Mary's Hospital, College of Medicine, The Catholic University of Korea, #56 Dongsu-ro, Bupyeong-gu, Incheon, Republic of Korea

Tel. +82-32-280-7369; Fax. +82-32-280-5349; email: doctorkwon@catholic.ac.kr

Round 2

Reviewer 1 Report

Comments and Suggestions for Authors

All issues have been addressed.

Reviewer 2 Report

Comments and Suggestions for Authors

The authors have been revising their manuscript according to comments from this reviewer.